# A Prospective, Blinded, Open-Label Clinical Trial to Assess the Ability of Fluorescent Light Energy to Enhance Wound Healing after Mastectomy in Female Dogs

**DOI:** 10.3390/ani14081250

**Published:** 2024-04-22

**Authors:** Andrea Marchegiani, Alessandro Troisi, Marilena Bazzano, Andrea Spaterna, Alessandro Fruganti

**Affiliations:** School of Biosciences and Veterinary Medicine, University of Camerino, 62032 Camerino, Italy; andrea.marchegiani@unicam.it (A.M.); alessandro.troisi@unicam.it (A.T.); andrea.spaterna@unicam.it (A.S.); alessandro.fruganti@unicam.it (A.F.)

**Keywords:** mastectomy, fluorescent light energy, wound healing, dog

## Abstract

**Simple Summary:**

Surgical wound management may represent a challenge due to possible complications and drug consumption. The present paper explores a novel fluorescence-based photobiomodulation (FLE) device as a wound management option in female dogs undergoing surgery for mammary cancer. Nine bitches received unilateral or bilateral mastectomy, and half of the wound was managed with FLE (a blue LED device that illuminates a roughly 2 mm layer of fluorescence-producing gel), while the remain part of the wound received no FLE. The illumination was repeated twice in the same session (one minute apart) and five days apart. Neither antimicrobials nor other drugs were administered to the dogs. All wound portions managed with fluorescent photobiomodulation showed a notably better quality of wound healing in terms of having fewer step-off borders, negligeable contour irregularities, and an absence of scar distortion. Moreover, when taking into account inflammatory indexes such as erythema, edema, and serous discharge, these were registered as being significantly lower for those wound portions illuminated with fluorescent photobiomodulation. The outcomes of this preliminary study underscore the positive impact of fluorescent photobiomodulation on the healing of post-mastectomy wounds in female dogs, with the chance to potentially replace certain topical treatments and improve the overall compliance of owners.

**Abstract:**

Mammary gland tumors represent the most frequently diagnosed malignant neoplasm in intact female dogs, and surgical removal represents the current gold standard treatment. To promote wound healing and prevent possible bacterial contamination, perioperative antimicrobials are commonly used in clinical practice, even though there are no publications establishing guidelines for the use of such drugs in canine mastectomy. The aim of the present study was to evaluate the ameliorative effect of fluorescent light energy on the quality of the healing process after mastectomy surgery in female dogs, in the absence of perioperative antimicrobial administration. Nine female dogs received a multiple-gland mastectomy due to gland tumors and received FLE application immediately after surgery and then five days after. The surgical incisions were evaluated by a blind investigator over time using the Modified Hollander Cosmesis and Modified Draize Wound Healing Score systems. Statistical analysis revealed a significant ameliorative effect of FLE in the control of step-off borders, contour irregularities, and excessive distortion. In addition, erythema, edema, and serous discharge were lower for those wounds managed with FLE. These results underscore the advantageous impact of FLE on the healing of post-mastectomy wounds in female dogs, offering the dual benefits of reducing potential infection risks and lessening the home care burden for pet owners.

## 1. Introduction

Mammary gland neoplasia represents one of the most common cancer diagnoses in female dog gynecology, with half of them being malignant in intact bitches [1]. Surgical removal of the lump and associated tissues represents the reference point therapy for such neoplasia in dogs, and is considered the sole intervention able to manage local cancer, apart from inflammatory carcinoma or the presence of distant metastases [2]. As another surgical procedure, mastectomy is not immune to complications. Post-surgical complications may include seroma formation, wound dehiscence and infection, skin necrosis, self-mutilation, edema of the hind limbs, and tumor recurrence [2,3].

Migration of normal cutaneous microbial flora is the most common route of surgical site infection (SSI) for many surgeries, including those related to mammary glands. Skin preparation, cleaning, and disinfection techniques aim to reduce or eliminate the risk of the transmigration of resident and transient bacterial flora at the wound site, thereby reducing the SSI rates and related the consumption of antimicrobial drugs [4].

Hair clipping and skin scrubbing are practices considered to be mandatory to reduce the risk of bacteria entering the surgical site and prevent surgical site infection (SSI) [5]. Despite the lack of both studies and a definitive consensus on the perioperative use of antimicrobial therapy, this practice is still overused in veterinary surgery [6,7]. Up-to-date international guidelines suggest that the implementation of antibiotic prophylaxis only in cases of effective clinical necessity and surgical routine interventions, including cutaneous and superficial soft tissue and clean abdominal incision techniques, is considered to not require antimicrobial administration [8]. Surgeries with no indication for antibiotic prophylaxis include routine procedures such as cutaneous and superficial soft tissue and clean abdominal incision techniques. The risk of developing an SSI largely depends on the extent of wound contamination with exogenous bacteria. Efficient pre-operative preparation of the patient’s skin and the used surgical technique can reduce the number of wound contaminants [8]. However, it must be recognized that clinical signs of infection can develop as long as 14 days postoperatively [9]. By this point, patients have often been discharged, and therefore, postoperative management must be achievable at home, by clients. A recent study identified SSIs in 2.83% of patients following discharge; no SSIs were detected prior to discharge, and the mortality rate of these patients was 8.3% after the development of sepsis [10].

In the last few decades, different research groups have started the exploration of non-pharmacological modalities to support wound care, providing dedicated scientific evidence [11,12,13,14]. For example, the application of hyperbaric oxygen therapy (HBOT) consists in administering a 100% oxygen mixture at increased pressure, to exert various biochemical effects, such as the improvement of antioxidant systems, modulation of the inflammatory response, antimicrobial activity, and others, improving the healing of different injuries and wounds, especially when an ischemic response and loss of tissue are taking place [15,16].

Another non-pharmacological intervention is represented by photobiomodulation (PBM), which consists of the administration of laser or light therapy at a non-thermal level to exert certain biological functions.

PBM is widely used in humans due to its benefits in tissue regeneration, the reduction in pain, and the control of inflammation [17,18]. PBM consists of the use of visible-to-infrared light to stimulate biological functions based on the presence of endogenous photo acceptors, acting as chromophores and widely expressed in different cell types, including those of the skin. PBM impacts biological processes, including inflammation, angiogenesis, and signal transduction pathways that recruit transcription factors activating several genes involved in multiple aspects of cell biology [19,20,21,22,23,24,25].

Fluorescent light energy (FLE) is a modality of PBM that uses chromophores to produce fluorescence and stimulates healing from different diseases, including surgical, uncomplicated, and orthopedics wounds [26]. In a previous study, wounds receiving FLE achieved complete re-epithelialization, less inflammation of the dermal layer, and the greater and more regular deposition of collagen, in comparison with control wounds. According to immunohistochemistry, the expression of factor VIII, epidural growth factor, decorin, collagen III, and Ki67 was increased in treated compared with untreated tissues [27]. To the best of the authors’ knowledge, only one case report on the use of chromophore gel-assisted blue light phototherapy for the treatment of surgical site infections in breast surgery [28] has been published, and in the context of human medicine, and no study has been conducted yet in the veterinary field.

The aim of the present study was to evaluate the ameliorative effect of FLE on wound healing after mastectomy surgery in female dogs. The study was designed to assess whether the combination of FLE with standard of care is able to improve the quality of healing process, assessed by a dedicated scoring systems, in the absence of pre-operative antimicrobial administration.

## 2. Materials and Methods

### 2.1. Ethical Considerations

This study protocol was approved by the Animal Welfare Body of the University of Camerino (protocol number 9/2021). Informed consent was obtained from all owners of the animals enrolled in the study.

### 2.2. Study Animals

Between April 2022 and September 2023, nine female dogs were presented to the Veterinary Teaching Hospital (VTH) for mastectomy procedures and met the criteria to be included in the study. Dogs amenable to hospitalization, with the agreement of their owners, were hospitalized in the VTH for five days after surgery under VTH standard protocols; the others were treated as out-patients. No experimental animals were used in this study. No restriction of age, breed, or bodyweights were considered in the animals recruited, but pregnant dogs were not enrolled in this study.

To be enrolled, the female dogs had to present at least a single or multiple palpable masses underneath the skin consistent with mammary tumors. The exclusion criteria were as follows: poor general health status based on physical examination performed by a veterinarian (on day 0) according to the Australian Veterinary Association Standard of Care Regular health check standards for dogs and cats; current administration of photosensitizing agents or products; current or previous history of systemic illness (including but not limited to diabetes mellitus, hypothyroidism, hyperadrenocorticism, growth hormone deficiency, leishmaniasis, and kidney malfunction); suspicion or presence of inflammatory mammary carcinoma; and solitary lesions smaller than 0.5 cm.

### 2.3. Study Protocol and Parameters Assessed

All procedures were compliant with standard veterinary practice for the diagnostic work-up and management of dogs with mammary tumors, and represent the routine approach adopted in VTH. Dogs underwent surgical removal of masses (either unilateral or bilateral mastectomy) only after the diagnosis of a mammary tumor.

Dogs underwent surgery after a food fasting period of 12 h; water was withheld for 12 h prior to surgery. Dogs were intramuscularly premedicated with butorphanol 0.04 mg/kg body weight (bw) and 0.02 mg/kg bw of acepromazine. Anesthesia was induced by injection of 5 mg/kg bw of propofol in an intravenous catheter placed in the left cephalic vein. Then, an endotracheal tube was applied, and anesthesia was maintained with an oxygen/isoflurane mixture. Post-surgical analgesia was provided by intravenous administration of butorphanol 0.02 mg/kg bw. No other medications (either antimicrobials or anti-inflammatory drugs) were administered.

In the fifteen minutes after surgery, during anesthesia recovery, dogs received the first application of the FLE regimen (Phovia, Vetoquinol, Lure Cedex, France), only in a half of the length of surgical incision. FLE application consisted of applying a rough 2 mm layer of gel and illuminating it with a blue LED device that delivers noncoherent blue light with a peak wavelength between 440 and 460 nm and a power density of between 55 and 129 mW/cm^2^, for 2 min, at approximately a 5 cm distance. After illumination, the gel was gently removed using sterile gauzes immersed in sterile saline solution and a second illumination performed, one minute apart. Depending on the case, a dry compressive bandage was applied and changed at regular intervals. FLE application was repeated five days after surgery (prior to discharging the patient for those hospitalized) using the same protocol (once daily).

The principal investigator assessed the dogs on day 0 (before surgery) and handed them over to the collaborating investigator for FLE application, remaining blinded to which part of the surgical incision received Phovia for the whole duration of the study.

Assessment visits were conducted by the collaborating investigator, who also performed treatments and uploaded photos of the lesions to an electronic shared folder to allow the principal investigator to blindly assess the surgical incisions using the Modified Hollander Cosmesis Score system and Modified Draize Wound Healing Score [29,30].

The Modified Hollander Scar Scale [29] items (Table 1) measure the relative severity of step-off borders (edges of wounds not on the same plane), contour irregularities (wrinkled skin near wound), margin separation (gap between sides of wound), and excessive distortion (edema and infection), each on a 5-point ordinal scale (0 = absence, 1 = trace, 2 = mild, 3 = moderate, and 4 = severe).

The Modified Draize scoring items (Table 2) [30] measure the relative severity of erythema (abnormal redness of the skin), edema (swelling caused by excess fluid), serous discharge (clear fluid present on the wound surface), and purulent exudate (pus present on the wound surface), each on a 5-point ordinal scale (0 = absence, 1 = trace, 2 = mild, 3 = moderate, and 4 = severe).

On day 0, two wound samples for culture and sensitivity testing were obtained: one, using a dry sterile swab, at the level of presumed surgical access immediately before the incision, and the second at the end of the surgery, immediately before completing the skin suture. To assess for surgical site infection, microbial culture and antimicrobial susceptibility testing were also carried out on days 3, 5 (before FLE application), and 7 post-surgery. At each time point, two wound samples were obtained for analysis—one from the control site and one from the FLE-treated site. The microbial culture and isolation methodology/antibiotic sensitivity of the microbiological samples were compliant with European Committee on Antimicrobial Susceptibility Testing (EUCAST) guidelines [31].

On day 0, and then on days 3, 5, and 7, the following assessments and procedures were carried out: general clinical examination; detailed examination of the skin and surgical incision; photographic documentation of the lesions. Moreover, in order to check for the presence of surgical site infection, on days 0 and 5, a neutrophil engulfing bacteria score (NES) in the range of 0–4 (Table 3) was obtained by pressing a clean microscope slide directly onto the surgical incision and staining it with a Romanosky (Diff quick) stain [32,33].

### 2.4. Statistical Analysis

GraphPad Prism version 8.2.1 for macOS (GraphPad Software, La Jolla, CA, USA) was used to perform the statistical analyses. Items from the modified Hollander Scar Scale (step-off border; contour irregularity; margin separation; excessive distortion) and Modified Draize scoring system (erythema; edema; serous discharge; purulent exudate) were analyzed individually, applying two-way analysis of variance (ANOVA) for repeated measures, considering treatment factor and time as sources of variation. In addition, Dunnett’s multiple-comparisons test was performed for each dataset, comparing the results obtained on days 3, 5, and 7 for each item with the one registered on day 0 (immediately after surgery). *p* values were considered significant when less than 0.05.

## 3. Results

The signalment data (breed, age, sexual status, type of mastectomy, and location of wound treated with FLE) of the dogs included in the study are reported in Table 4, as well as the anatomopathological diagnosis of mammary neoplasia.

The results from Modified Hollander Cosmesis Score system and Modified Draize Wound Healing Score are reported in Figure 1 and Figure 2.

The statistical analysis revealed a significant effect of wound management (Figure 3) on the presence of step-off borders (*p* = 0.0003), contour irregularities (*p* = 0.0127), and excessive distortion (*p* = 0.0262) in favor of FLE. The same effect was found for erythema (*p* = 0.0384), edema (*p* = 0.0048), and serous discharge (*p* = 0.0015).

Dunnett’s multiple-comparisons tests revealed significant differences between the FLE and control wounds for all the items from the Hollander and Draize scoring systems, except margin separation and purulent discharge, as reported in Table 5 and Table 6.

The microbiological swabs performed before the incision of the skin revealed the predominant presence of *Staphylococcus* spp. (n = 9); the isolated species were *S. pseudintermedius* (9) and *S. aureus* (3). Other bacteria such as *Streptococcus* spp. and *Enterococcus* spp. were also detected, but less frequently. No bacteria were found to be present at the time of second sampling, at the end of the surgery, and immediately before completing skin suture.

Microbiological swabs were also obtained 3, 5, and 7 days after surgery. In the control wounds, we detected bacteria in 4, 3, and 1 dog on day 3, day 5, and day 7, respectively; FLE-managed wounds were found to be negative for bacteria. The isolated bacterial strains for each type of wound and timepoint are reported below in Table 7.

All dogs included in this study received a multiple-gland mastectomy.

Neutrophils engulfing bacteria were only found in dog 8, on days 3 and 5, when purulent exudate was present, and only in control-managed portion of the wound. No other dogs revealed the presence of neutrophils engulfing bacteria.

## 4. Discussion

This prospective randomized clinical trial aimed to evaluate the effect FLE on enhancing wound healing post-mastectomy in female dogs. The improvement in wound healing and the quality of surgical incisions were substantiated through clinical examinations and quantitatively assessed using the Modified Hollander Cosmesis Score system and Modified Draize Wound Healing Score. The findings from this study indicate that FLE, in conjunction with standard of care, significantly stimulated the wound healing process and ameliorated the macroscopic appearance of surgical incisions when compared to the control sites.

This was evidenced by statistically significant lower scores for step-off borders, contour irregularities, excessive distortion, erythema, and edema when compared to the control sites. Furthermore, the FLE-treated wounds registered a notable decrease in the purulent exudate and margin separation scores compared to the control site.

This outcome is opposite to that of Gammel and collaborators [34], who found no beneficial effect of low-level light therapy (LLLT) after flank ovariectomy in dogs. The authors administered LLLT once daily for 5 days using a 980 nm laser and a total energy density of 5 J/cm^2^ to dogs undergoing bilateral flank ovariectomy (each dog represented a control of herself, as in the present study) and the wounds were evaluated 3, 7, 11, and 14 days after surgery. The authors found no difference between groups for the subjective assessment of healing time and wound measurements. Similarly, and except for control lesions that showed more necrosis and perivascular lymphocytes and macrophages seven days after surgery, no difference in histopathologic assessment was found, and the authors concluded that LLLT did not appear to influence the healing of surgically created incisions and small wounds [34].

Although Gammel and collaborators did not recommend LLLT to stimulate the healing of uncomplicated small wounds and incisions, the results of the present study support the management of mastectomy wounds with FLE.

The enhanced macroscopic and aesthetic appearance of wounds treated with FLE can be attributed to its positive influence on the biological and cellular mechanisms that underpin the healing process. Previous research has demonstrated that wounds subjected to FLE treatment exhibit complete re-epithelialization, diminished inflammation of the dermal layer, and a more substantial and consistent deposition of collagen relative to control wounds [27]. Immunohistochemical analyses further support these findings, revealing an elevated expression of factor VIII, epidermal growth factor, decorin, collagen III, and Ki67 in tissues treated with FLE compared to those that were untreated, underscoring the pivotal role of FLE in modulating the key factors involved in wound repair and regeneration (15).

In the present study, antibiotics were not administered peri-surgically, yet no infections or purulent discharges were noted in areas treated with FLE. Notably, only one dog developed self-limiting purulent discharge, which was confined to the area of the wound that had not been treated with FLE. These findings are in accordance with the surgical site infection (SSI) rate reported for mastectomy in dogs [35].

In the study of Spare and collaborators, the SSI incidence in a population of bitches that underwent excisional surgery for mammary gland neoplasia without perioperative antimicrobial prophylaxis was lower than or similar to previously reported incidences of SSIs in dog populations that have undergone tumor excisional surgery plus perioperative antibiotics [5]. In addition, the authors found that the excision of two or more glands represents an increased risk of developing SSI and non-SSI complications compared to dogs that had one gland excised, and also found that increased body weight is associated with an increased risk of non-SSI complications. Spare and collaborators concluded that the routine use of perioperative antibiotics in tumor excisional surgery can be questioned, at least in single-gland mastectomy, in otherwise clinically healthy dogs [5].

In our study, microbiological swabs collected prior to surgical incisions and on days 3, 5, and 7 post-operation demonstrated a significant difference in bacterial colonization between sites treated with FLE and the control sites, particularly concerning common pathogens such as *Staphylococcus* spp., *Streptococcus* spp., *Enterococcus* spp., and *Pseudomonas* spp. The absence of bacteria in wounds treated with FLE is particularly noteworthy. These observations are consistent with the existing literature on FLE, suggesting that this intervention may exhibit antimicrobial effects, either directly by impacting bacterial cells or indirectly by bolstering the host’s innate immune response and altering the wound microenvironment to deter bacterial survival and proliferation.

These findings indicate that FLE could serve as a valuable adjunct in surgical wound management, aiming to reduce colonization or infection risk, especially in contexts where the use of antibiotics is to be minimized due to concerns over resistance or adverse effects.

As demonstrated by Elishar [9], clinical signs of infection can emerge up to 14 days post-surgery, a time when most patients have usually been discharged. In a veterinary study by Turk and collaborators [36], active post-discharge surveillance identified nine cases of surgical site infections (SSIs), constituting 34% of the total, which would have remained undetected without such surveillance efforts. Consequently, it is advisable for practitioners to establish a post-discharge surveillance program tailored to specific procedures.

Implementing the FLE protocol necessitates pet owners to return to the clinic for procedure applications. This facilitates recheck visits, ensuring thorough postoperative monitoring.

In the present study, FLE was applied with a five-day interval to reflect the real-life clinical setting where owners are required to return to the clinic for rechecks. Although this FLE protocol is different from those previously described (once weekly back-to-back or single illumination twice weekly), it has been demonstrated to be effective in promoting wound healing and preventing bacterial colonization. Such structured follow-ups inherently reduce the likelihood of infections going unnoticed, as they allow for the timely detection and management of potential complications, thereby potentially lowering the risk of undetected infections. Additionally, FLE may supplant some topical treatments, enhancing adherence to home care regimens and easing the management of post-surgical therapies.

Despite the promising results presented in this study, some limitations must be acknowledged to contextualize our findings within the broader spectrum of veterinary medicine and postoperative care research. First, the sample size of this trial was relatively small, which may limit the generalizability of the results to the wider population of female dogs undergoing mastectomy. A larger cohort would enhance the statistical power of the study and provide a more robust evaluation of FLE’s efficacy. Second, the study design did not include long-term follow-up assessments beyond the immediate postoperative period. Consequently, the durability of the observed beneficial effects of FLE on wound healing and its potential impact on long-term outcomes, such as scar formation, dehiscence, and infection, remain uncertain. Future studies incorporating extended follow-up periods are necessary to fully ascertain the long-term benefits and potential drawbacks. Additionally, this study was conducted within a single veterinary institution, which may introduce institutional biases related to surgical techniques, postoperative care practices, and patient management protocols.

Recent developments in advanced wound care technology in veterinary medicine include different healing-enhancing techniques, such as laser therapy or hyperbaric oxygen therapy, to increase wound healing outcomes with minimal side effects. As a supplementary limitation, the study herein presented explored just FLE as a technique that could enhance healing, and future investigations would benefit from supplementary groups managed with these next-generation innovative approaches.

Multi-center trials including parallel groups treated with different nonpharmacological healing-enhancing techniques would help to mitigate these biases and validate our findings across different settings and populations.

## 5. Conclusions

In conclusion, the outcomes of this trial underscore the advantageous impact of FLE on the healing of post-mastectomy wounds in female dogs, as evaluated by various assessment tools. This underscores its value as a significant adjunct to conventional postoperative care in veterinary medicine, offering the dual benefits of reducing potential infection risks and lessening the home care burden for pet owners. FLE’s application could potentially replace certain topical treatments and improve overall compliance by simplifying the administration of home therapies, thereby relieving pet owners of some responsibilities associated with postoperative care.

## Figures and Tables

**Figure 1 animals-14-01250-f001:**
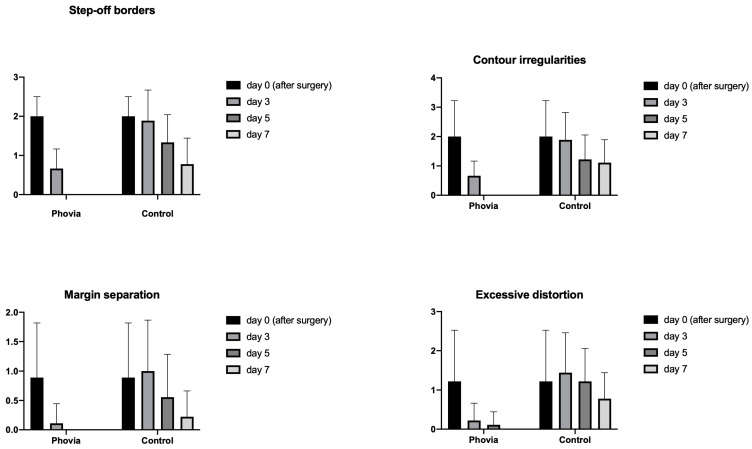
Modified Hollander Cosmesis Score item variation over time and in each type of wound management.

**Figure 2 animals-14-01250-f002:**
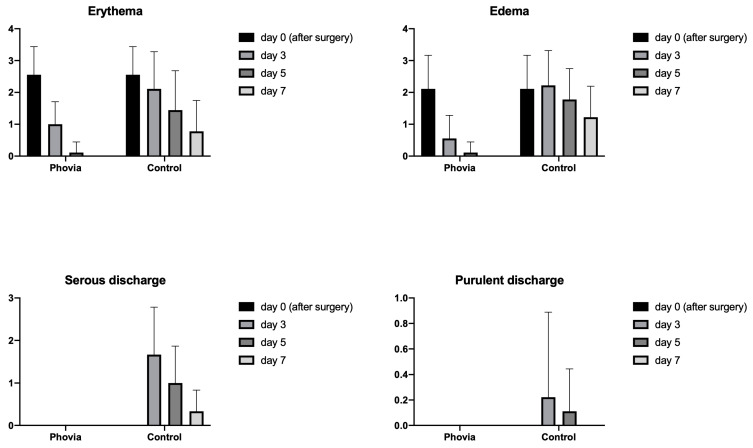
Modified Draize Score item variation over time and in each type of wound management.

**Figure 3 animals-14-01250-f003:**
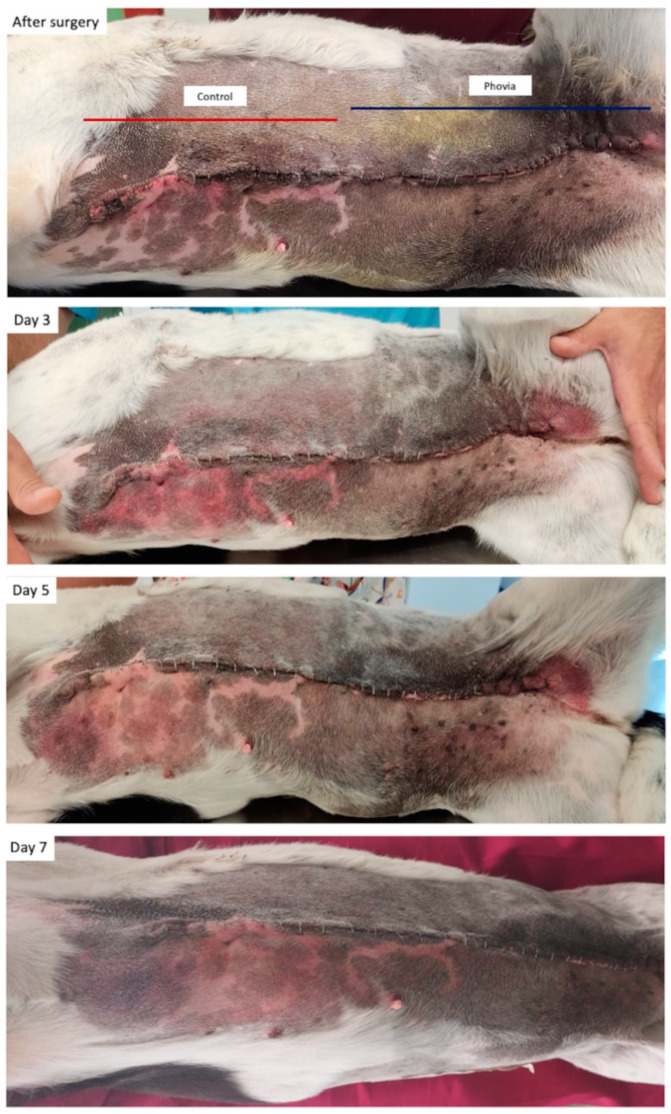
Dog (#7) included in the study.

**Table 1 animals-14-01250-t001:** Modified Hollander Scar Scale.

Score	Step-Off Borders	Contour Irregularity	Margin Separation	Excessive Distortion
0	No step-off borders	No contour irregularities	No edges of wound are apparent, looks like normal skin	No edema or appearance of infection
1	Very slight borders (barely perceptible)	Very slight wrinkling (barely perceptible)	Very slight distance between wound edges (barely perceptible)	Very slight amount of edema and indication of infection
2	Well-defined step-off borders	Slight wrinkling around wound	Slight but apparent distance between wound edges	Slight but apparent amount of edema and indication of infection
3	Moderate-to-severe step-off borders	Moderate wrinkling around wound	Moderate distance between wound edges, but less than original wound size	Moderate amount of edema and indication of infection
4	Severe step-off borders (wound edges are on very different planes from one another)	Severe wrinkling around wound	Maximum observable distance between wound edges as upon creation of wound	Severe amount of edema and indication of infection

**Table 2 animals-14-01250-t002:** Modified Draize scoring system.

Score	Erythema	Edema	Serous Discharge	Purulent Exudate
0	No erythema	No edema	Wound is dry	No purulent exudate
1	Very slight (barely perceptible)	Very slight (barely perceptible)	Very slight (barely perceptible)	Small amount of purulent exudate—no color
2	Well-defined erythema	Slight edema (edges of area well defined by definite raising)	Slight serous discharge	Moderate amount of purulent exudate—no color
3	Moderate-to-severe erythema	Moderate edema (raised approximately 1 mm)	Moderate serous discharge with blood-tinged fluid	Moderate amount of purulent exudate—red- or green-tinged color
4	Severe (beet redness)-to-slight eschar formation	Severe edema (raised more than 1 mm and extending beyond area of exposure)	Large volume of serous discharge with marked blood-tinged fluid	Purulent exudate including accumulation in subcutaneous tissues at wound margin—abscess formation

**Table 3 animals-14-01250-t003:** Severity of neutrophil engulfing bacteria scores (NES).

NES Score	Numbers of Neutrophils Engulfing Bacteria *
0	None seen
1	<1
2	1–4
3	5–10
4	>10

***** Numbers of neutrophil engulfing bacteria per high powered field (×500 magnification), average on 10 microscopic fields.

**Table 4 animals-14-01250-t004:** Study participants.

#	Breed	Age (Years)	Sexual Status at VTH Presentation	Neoplasia	Type of Mastectomy	Location of FLE Treatment *
1	Mixed breed	12	neutered	solid carcinoma	unilateral	cranial
2	White Swiss shepherd dog	7	intact	tubular adenocarcinoma	bilateral	caudal
3	Lagotto Romagnolo	6	intact	solid carcinoma	unilateral	cranial
4	Italian bloodhound	9	intact	tubular adenocarcinoma	unilateral	cranial
5	Springer spaniel	6	intact	tubular adenocarcinoma	bilateral	caudal
6	White Swiss shepherd dog	8	intact	tubular adenocarcinoma	bilateral	caudal
7	Mixed breed	13	intact	tubular adenocarcinoma	bilateral	caudal
8	English setter	10	intact	solid carcinoma	bilateral	caudal
9	German shepherd	8	intact	solid carcinoma	bilateral	cranial

* the counterpart of the surgical wound served as control and received no illumination or other interventions.

**Table 5 animals-14-01250-t005:** Dunnett’s multiple-comparisons test results for Modified Hollander Cosmesis Score.

	Step-Off Borders		Contour Irregularities		Margin Separation		Excessive Distortion	
	*p* Values	Summary	*p* Values	Summary	*p* Values	Summary	*p* Values	Summary
FLE								
day 0 (after surgery) vs. day 3	0.0001	***	0.0044	**	0.0202	*	0.0419	*
day 0 (after surgery) vs. day 5	<0.0001	****	0.0031	**	0.0504	ns	0.0517	ns
day 0 (after surgery) vs. day 7	<0.0001	****	0.0031	**	0.0504	ns	0.0549	ns
*Control*								
day 0 (after surgery) vs. day 3	0.9491	ns	0.9491	ns	0.9681	ns	0.7681	ns
day 0 (after surgery) vs. day 5	0.1165	ns	0.0563	ns	0.6438	ns	>0.9999	ns
day 0 (after surgery) vs. day 7	0.0138	*	0.0504	ns	0.054	ns	0.5947	ns

* *p* < 0.05; ** *p* < 0.01; *** *p* < 0.001; **** *p* < 0.0001; ns = not significant.

**Table 6 animals-14-01250-t006:** Dunnett’s multiple-comparisons test results for Modified Draize Score.

	Erythema		Edema		Serous Discharge		Purulent Discharge	
	*p* Values	Summary	*p* Values	Summary	*p* Values	Summary	*p* Values	Summary
FLE								
day 0 (after surgery) vs. day 3	<0.0001	****	<0.0001	****	-		-	
day 0 (after surgery) vs. day 5	<0.0001	****	0.0003	***	-		-	
day 0 (after surgery) vs. day 7	<0.0001	****	0.0008	***	-		-	
*Control*								
day 0 (after surgery) vs. day 3	0.3573	ns	0.9776	ns	0.0053	**	-	
day 0 (after surgery) vs. day 5	0.0177	*	0.5485	ns	0.0213	*	-	
day 0 (after surgery) vs. day 7	0.0001	***	0.0056	**	0.1829	ns	-	

* *p* < 0.05; ** *p* < 0.01; *** *p* < 0.001; **** *p* < 0.0001; ns = not significant. For serous discharge in FLE group and purulent discharge in both groups, it was not possible to perform statistical evaluation due to the paucity of data for such items.

**Table 7 animals-14-01250-t007:** Frequency of isolated bacteria from microbiological swabs.

	Enrolment		Day 3		Day 5		Day 7	
Isolated Bacteria	before Surgery	after Surgery	FLE	Control	FLE	Control	FLE	Control
*Staphylococcus pseudintermedius*	9	-	-	3	-	2	-	1
*Staphylococcus aureus*	3	-	-	4	-	1	-	-
*Streptococcus* spp.	2	-	-	1	-	1	-	-
*Enterococcus* spp.	1	-	-	-	-	1	-	1
*Pseudomonas* spp.	-		-	1	-	1	-	-

## Data Availability

The data presented in this study are available on request from the corresponding author.

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
