# Peer review of "A Prospective, Blinded, Open-Label Clinical Trial to Assess the Ability of Fluorescent Light Energy to Enhance Wound Healing after Mastectomy in Female Dogs"

_animals, 2024, doi:10.3390/ani14081250_

Round 1

Reviewer 1 Report

Comments and Suggestions for Authors

The study described in this article is an evaluation of the effect of the use of a fluorescent light beam on the skin in the surgical incisions of bitches undergoing radical mastectomy. Half of the incision is set as the control (untreated) area versus the other half of the incision of the same animal. Only 9 animals were finely included in the study. It is concluded that the use of this technique may reduce the incidence of surgical site infection and improve the surgical wound healing process.

  •  
  •  

The main strength of this study is that there are no published studies showing these results. The main weakness is that the number of animals included in the study is limitedThe methodology used to obtain scientific evidence of the light effect seems to me to be correct and the use of a blinded assessor who does not know which of the two sections of the incision has been treated allows a comparative assessment between the control area and the treated area.

On the other hand, the statistical study appears to be correct to the best of my knowledge.

The graphs add visual value to the results.

The tables are not adapted to the requirements of the journal, so we recommend following the instructions of the authors in order to adapt them to the requirements in the final edition, if accepted.

The conclusions are supported by the observations of the study itself.

The limitations of the study have been included, highlighting that the number of animals used is very limited and that there may be deviations due to the specific conditions of the hospital where the study is performed.

Also, the size of the oncological lesion is missing, as the extent of the mastectomy at the site where the lesion was located could add a difference between patients.

Appropriate permissions are included for a non-experimental clinical procedure.

Specific comments.

Line 11: represents instead of represent?

Line 76: if finally only 9 patients were included it is not necesary to detail this 4 patients. In any case, if you leave these 4 patients you should describe why they are included as part of the study if they are not used.

Line 96: please details the number of unilateral versus bilateral mastectomy included in this study

Line 96 and 97: Please add name of drugs used and doses. Anaesthesic protocol is missing.

It is not clear if antibiotic and analgesic treatments were used during hospitalization and after discharge. In that case, please include details.

The description of the culture and isolation methodology and antibiotic sensitivity of the microbiological samples is missing in Material and methods. There are no antibiotic sensitivity/resistance results. This is important because it is referred to in the conclusions.

In Figure 3. It is clear that in this patient the caudal region was used as the treatment area. Please describe in the main text (Results) how many patients were treated in the caudal region and how many in the cranial region. Both regions have different possibilities to be sites of surgical infection due to their anatomical position.

Line 229-230. You cannot conclude these results if they are not supported by statistically significant differences. Use an appropriate expression, such as apparently, but you cannot be so explicit and then say that there were no statistically significant differences.

Line 233 correct the mistake “found who found no beneficial effect”

Line 249: Add the references to those previous research at the end of the sentence.

Line 270: correct the mistake “that that”

Author Response

Dear Reviewer, authors of the present paper would sincerely thank you for the appreciation of the work done and for the insights to improve the quality of the paper. We have carefully taken into account your recommendations, which has been promptly addressed. Please find in bold blue color the specific answer to the points raised.

Reviewer 2 Report

Comments and Suggestions for Authors

Introduction:

Line 29: the first sentence is a generalist perspective, it should be supported with more references.

Line 32: it should be supported with more references

Line 39 and 47: preoperative antimicrobial therapy?? what is the standard? what is the guideline used according with the preoperative surgery time?

Line 46-47: "efficient pre-operative preparation of the patient´s skin preparation"? what is the normal? explain and give this information.

Line 53: " Several studies have explored the ability of fluorescent light energy (FLE) on the healing from different diseases [9–16]" - these are studies from the same research team. You have to present other studies. If there isn´t other studies, since these is pioneer team, you have to refer that there is other modalities that already showed scientific evidence, like low-level-laser-therapy and laser class IV. It is also important to refer the application of hyperbaric oxygen therapy (HBOT), that improves healing of different injuries and wounds, when they show ischemic and loss of tissue. Besides the reference of other modalities, you should also find references in human medicine

Line 61: If there isn´t any study, you have to refer the adjuvant modalities, and also the standart treatment for wounds, like hydrogel, coloid, paraffin compresses, honey action, and also ozone ointments.

Materials and methods:

line 71: refer that it was approved by ethic committees, and refer its name;

Line 72:  indicate the time for collecting the 13 females, since this is not an experimental study, it is in clinical environment.

Line 95: "dogs underwent surgical removal of masses , regional, unilateral or bilateral". There is a major variability between healing processes, when comparing approaches with different healing requirements. A total mastectomy has more predisposition to infection, since it is an extensive skin lesion.

Line 96: which opioids? which suitable drugs and non-steroidal anti-inflammatory therapy? Was it always the same pharmacological management?

Line 99: "during anesthesia recovery" it is unspecific, was it at which time? Did all the female applied in the same time? 

Line 103: Image of FLE application;

Line 106: "compressive bandage", without any other application of topical ointments, or hidrogel, cooled, etc?

Line 107: "was repeated five days", but once a day, twice a day?

Discussion:

Line 226 - you should explain differently...Will the FLE regimen influence the adjacent healing tissues? How can we compare truly without a control group?

On discussion you have to add more references, about laser therapy, since there is also positive results regarding LLLT.

To prove something you should have add a group with laser therapy and another with FLE, therefore you have to discuss with more references, for example:

- https://www.ncbi.nlm.nih.gov/pmc/articles/PMC8837844/

- https://www.ncbi.nlm.nih.gov/pmc/articles/PMC6362418/

- https://www.ncbi.nlm.nih.gov/pmc/articles/PMC8448658/

Search more references.

You haven´t refer any HBOT studies. You have to discuss, review and complete all the discussion

You only have one laser reference..

Attention to the references, because it the are very few references and some are repeated: reference 10, is the same of 12 and 21; and reference 11 is the same of 15.

You need major changes in the manuscript struturation, references and more important in discussion

Author Response

Dear Reviewer, authors of the present paper would sincerely thank you for points raised during revision process, with the aim to improve the quality of the paper. We have carefully taken into account your recommendations, which has been promptly addressed. Please find in bold blue color the specific answer to the points raised.

Round 2

Reviewer 2 Report

Comments and Suggestions for Authors

After this major revision, the manuscript was much improved and all the authors could respond and cope with all the suggestions made.

The only thing that I still suggest for them to do is at the end of the discussion, the authors should state the study's limitations. One particular limitation is that this study does not compare with other modalities that could enhance healing, such as laser therapy or hyperbaric oxygen therapy.

In my opinion, after this minor change, the manuscript is ready for publishing. 

Author Response

Dear Reviewer,

Authors would thank you for your time and expertise. We have expanded the study limitations’ section adding a proper sentence (lines 423-428) regarding the lack of comparison of the FLE technique with other healing techniques as such as laser therapy or hyperbaric oxygen therapy.